# Alzheimer’s Disease: Significant Benefit from the Yeast-Based Models

**DOI:** 10.3390/ijms24129791

**Published:** 2023-06-06

**Authors:** Khoren K. Epremyan, Dmitry V. Mamaev, Renata A. Zvyagilskaya

**Affiliations:** A.N. Bach Institute of Biochemistry, Research Center of Biotechnology of the Russian Academy of Sciences, Leninsky Ave. 33/2, 119071 Moscow, Russia; k.epremyan19@gmail.com (K.K.E.); dmamaev_inbi@mail.ru (D.V.M.)

**Keywords:** Alzheimer’s disease, yeast models, amyloid-β peptide, phosphorylated tau protein, mitochondrial dysfunction, mitotherapy

## Abstract

Alzheimer’s disease (AD) is an age-related, multifaceted neurological disorder associated with accumulation of aggregated proteins (amyloid Aβ and hyperphosphorylated tau), loss of synapses and neurons, and alterations in microglia. AD was recognized by the World Health Organization as a global public health priority. The pursuit of a better understanding of AD forced researchers to pay attention to well-defined single-celled yeasts. Yeasts, despite obvious limitations in application to neuroscience, show high preservation of basic biological processes with all eukaryotic organisms and offer great advantages over other disease models due to the simplicity, high growth rates on low-cost substrates, relatively simple genetic manipulations, the large knowledge base and data collections, and availability of an unprecedented amount of genomic and proteomic toolboxes and high-throughput screening techniques, inaccessible to higher organisms. Research reviewed above clearly indicates that yeast models, together with other, more simple eukaryotic models including animal models, *C. elegans* and *Drosophila*, significantly contributed to understanding Aβ and tau biology. These models allowed high throughput screening of factors and drugs that interfere with Aβ oligomerization, aggregation and toxicity, and tau hyperphosphorylation. In the future, yeast models will remain relevant, with a focus on creating novel high throughput systems to facilitate the identification of the earliest AD biomarkers among different cellular networks in order to achieve the main goal—to develop new promising therapeutic strategies to treat or prevent the disease.

## 1. Introduction

Alzheimer’s disease (AD) is associated with the name of Alois Alzheimer, a German neurologist who first revealed plaques and tangles in the cerebral cortex cells of a woman suffering from years of language dysfunction and memory problems, which, in his opinion, was beyond typical dementia [1]. However, the molecular compounds underlying pathology and qualitatively described by Alzheimer, insoluble β-amyloid peptide of plaques and truncated and hyperphosphorylated tau protein of intraneuronal inclusions (neurofibrillary tangles, NFTs), were identified only in the mid-1980s [2,3,4]. Later, the deposition of both extracellular senile plaques and NFTs, loss of synapses and neurons, and changes in the morphology and function of microglia and astrocytes were recognized to be the most pathologically important phenotypic hallmarks of Alzheimer’s disease (AD) [5,6]. Recent studies using synaptic vesicle glycoprotein 2A (SV2A) positron emission tomography (PET) imaging confirmed a significant association between synaptic density and cognitive abilities and suggested that this correlation is characteristic of the early stages of the disease [7].

AD can be the early onset familial autosomal dominant inherited (FAD) form (1–2% of all AD cases) or the most prevalent late-onset “sporadic” (SAD) form. FAD is mainly associated with mutations in the amyloid precursor protein (APP) and presenilin genes PSE1 and PSE2, significantly increasing the production of the most toxic fragment, Aβ42 [8,9]. AD is a long-developing, multifactorial, complex (involving genetic, environmental, metabolic, and other factors), age-dependent neurodegenerative disorder, and it is the predominant form of dementia, currently accounting for 70–80% of all late-life dementia cases and destroying the lives of over 50 million people [10,11,12,13]. AD is recognized by the World Health Organization as a global public health priority and as one of the most devastating diseases humanity has faced in the 21st century [14].

Aβ peptides are generated in both neurons and astrocytes from the amyloid precursor protein (APP) by successive secretase-mediated cleavage. As a result, Aβ peptides containing from 38 to 43 amino acids are formed, normally mainly Aβ40, more rarely the longer Aβ42. A rebalancing between Aβ40 and Aβ42 in favor of the Aβ42 form, which is highly amyloidogenic hydrophobic, more inclined to aggregation, and toxic, is a prerequisite for the development of AD. At the same time, Aβ aggregation is facilitated by its isomerization [15]. Tau protein, produced by the alternative splicing of the *MAPT* gene, is predominantly present in neuronal and glial axons, where it is vitally important for many physiological processes. It is highly charged and hydrophilic, however, in AD, hyperphosphorylated (P-Tau) tau monomers tend to aggregate first in oligomers and then in neurofibrillary tangles [16] (for details, see Section 2.2).

Several risk factors can lead to AD; among these, the top priorities are aging and immunosenescence [17], while apolipoprotein E4 and mutations in other cholesterol transporters in the brain are the strongest genetic risk factors [10,18,19,20]. Additionally, dozens of variants have been implicated in AD risk through large-scale Genome-Wide Association Studies (GWAS), which has become a well-recognized powerful method for finding genomic areas underlying observed phenotypic variation. A GWAS study of AD reported 44 single-nucleotide polymorphisms (SNP) associated with the late-onset AD [21]. The precise effects of most of these variants, however, have yet to be determined [22].

Other risk factors are considered as modifiable. These include disorders of the blood–brain barrier [23,24], vascular disorders [25], metabolic factors [26], gender (female) predominance [27,28,29], infections, inflammation [30], lifestyle factors, dysbiosis [31], family history [32], and others.

The “amyloid cascade hypothesis”, postulating that excessive accumulation of Aβ aggregates promotes tau protein hyperphosphorylation, causing destabilization of transport in axons and thus synaptic dysfunction; activation of astrocytes and microglia, causing the formation of cytokine-mediated reactive oxygen species (ROS); and inflammation causing oxidative mitochondrial damage, mitochondrial dysfunction, release and activation of pro-apoptotic factors, neuronal apoptosis leading to synaptic failure and neuronal loss and neurotoxicity and dementia, was dominant for the last three decades [33,34,35]. However, the failure of the development of effective drugs for reversing or slowing down the progression of AD requires innovative thinking and therapeutic approaches [13,36,37,38]. In fact, in AD patients, the loss of synapses and neurons does not always correlate with deposition of amyloid plaques [39,40], and cognitive deficits appeared decades before insoluble amyloid plaque deposition was clinically diagnosed [41,42].

Now it becomes clearer that AD, as well as numerous other neurodegenerative diseases, is multifactorial in nature, being a combination of genomic, epigenomic, metabolic, environmental, and other factors. Moreover, the complicated interactions of disease mechanisms and homeostatic biological networks will determine the early manifestation rate of development of the disease, measured by objectively monitoring pathology parameters in patients throughout their life [27,43].

Accurate early diagnosis of disease using sensitive, specific, non-invasive biomarkers is now a priority in AD research [44,45].

Researches have come to the realization that complex multifactorial diseases, such as AD, can be treated under systems biology studies—“first, in simple disease models such as yeast, aimed at dissecting the interplay of many factors involved, with subsequent studies in human to validate the predictions made using these simple models” [46].

## 2. Yeast as a Model of AD

The crucial need for a deeper understanding of the molecular mechanisms underlying AD, with its diversity of symptoms and sophisticated cross-talk of cofactors, has forced researchers to focus their attention on well-defined single-celled yeasts, the simplest eukaryotic organisms [47]. In fact, the *Saccharomyces cerevisiae* genome was the first eukaryotic genome to be fully sequenced in 1996 [48]. Now, plenty of genome-wide data have become readily available in comprehensive databases. It has been estimated that nearly sixty percent of yeast genes are homologous to human genes or have at least one conserved domain, and approximately thirty percent of all human gene orthologues can be found in *S*. *cerevisiae* [49,50]. The deletion [51] and overexpression [52,53] libraries were created, allowing for determining gene function and phenotypic analysis of mutants. *S. cerevisiae* was also the first organism where genome-wide transcriptional profiling was carried out, using cDNA microarrays [54].

Yeast is particularly useful for high-performance screening studies that are not achievable for humans. Estimation of thousands of phenotypes and testing of hundreds of molecules and targeted genes is greatly facilitated by the use of highly automated screening technologies and the available deletion and hyperexpression libraries [55,56].

Thus, yeasts, as an experimental model system, offer great advantages over other disease models due to the simplicity, short life cycle, low-cost cultivation techniques, relatively simple, cheap, and quick genetic and environmental manipulations, the large knowledge base and data collections, and availability of an unprecedented number of genomic and proteomic toolboxes and high-throughput screening techniques (Figure 1).

*S. cerevisiae* shows a high degree of conservation of different basic biological processes with all eukaryotic cells, such as gene interactions and recombination [57], the regulation of the cell cycle [58], organelle function, energy metabolism [59], mitochondria biogenesis [60], protein folding, quality control and degradation [61], proteostasis network [62], endocytosis [63], secretion [64], vesicular trafficking [65], autophagic pathways [66], programmed cell death [67], and many key signaling pathways, such as mitogen-activated protein kinase (MAPK) [68] and target of rapamycin (TOR) [69]. Moreover, in the majority of cases, these pathways were originally identified and studied in yeasts.

Collectively, all these data make it clear why the yeast *S. cerevisiae* is a powerful model eukaryotic cell for studying the fundamental cellular processes and protein functions that are also associated with much more complex multicellular eukaryotes such as humans and why this organism has become a valuable tool to unravel the molecular basis and features of human neurodegenerative disorders, including AD.

However, neurons have high energy requirements, relying on mitochondrial oxidative phosphorylation. Thus, the facultative aerobic yeast *S. cerevisiae*, with its few, small, and poorly structured mitochondria, cannot be considered the bioenergetic equivalent of neurons. Therefore, in addition to well-established *S. cerevisiae* models, new, upcoming, so-called non-convential yeasts *Schizosaccharomyces pombe*, *Candida glabrata*, *Kluyveromyces lactis*, and *Yarrowia lipolytica* yeast models for AD were described [70,71]. *Y. lipolytica*, an aerobic “multitalented” yeast species with a well-described genome, has GRAS (generally regarded as safe) status; the versatile substrate utilization profile, rapid utilization rates, metabolic diversity and flexibility, unique biosynthetic and secreting capacities, and energy metabolism that are closely reminiscent of that in mammals, susceptible to molecular genetic engineering tools, and having a long history of using heterologically expressed proteins due to unique biosynthesis and secreting capacities (reviewed by [72]) may be an extremely promising alternative model for deciphering mitochondria-associated AD pathogenesis. Moreover, the *Y. lipolytica* yeast with an endotoxin-free host system has some additional advantages for recombinant protein production [73].

For a long time, the contribution of yeast to clarify mechanisms underlying AD in humans was limited to genes having homologues in yeast. Recently, however, the so-called “humanized yeast model systems”, where heterologous expression of human genes occurs without a yeast homologue, are becoming implemented as an essential research instrument for determination of molecular mechanisms underlying human disease [13,71,74]. However, it should be noted that despite being a simplified and powerful model system, yeasts, being unicellular organisms, have some evident limitations in application to neuroscience. Multicellularity, cell–cell interactions, synaptic transmissions, axonal transport, glial–neuronal interplay, immune and inflammatory responses, cognitive aspects of AD, and other neuronal specializations cannot be recapitulated in yeasts. Despite obvious limitations, yeast models, when used early and complementary to mammalian systems, can significantly contribute to a better understanding of sophisticated cross-talk of AD-related cellular processes such as oxidative stress, mitochondrial dysfunction, disordered proteostasis, autophagy, and others and serve as a powerful first-line screening system for identification of novel targets and novel agents against neurodegeneration.

Several excellent reviews have recently dealt with specific aspects of AD using yeast models [13,62,75,76,77,78]. However, a lack of effective AD therapy, despite tremendous efforts in elucidating the molecular and cellular players involved in AD pathology, the resisted idea of dementia and AD as a fate [79], the growing hope of researchers, and the disappointment from the public related to finding effective means to combat AD by using different approaches, including models, among them yeast-based, has motivated us to write a new review to sum up what has been newly discovered since the last AD reviews and share our thoughts on how best to mitigate or to slow down AD progression.

In the following sections, we will discuss in more detail the contribution of yeast-based models to understanding how amyloidogenic proteins are formed, how toxic they are, and what means are already found to prevent their formation.

### 2.1. Yeast as a Model of Aβ Generation, Aggregation, Toxicity, and De-Aggregation

#### 2.1.1. Yeast as a Model of APP Processing

Since the description more than two decades ago of the pioneering yeast (*S. cerevisiae*) AD model, so-called “humanized yeast model systems” heterologically expressing mammalian Aβ and tau have been constructed for gaining new insights into APP in vivo processing, Aβ localization, oligomerization, and toxicity.

Aβ peptides are generated by both neurons and astrocytes from the amyloid precursor protein (APP), a type I single-pass transmembrane protein present at high levels with expected functions associated to signaling, cell adhesion, and neuronal maturation and migration [80]. After production in the endoplasmic reticulum (ER), the protein is then transferred from the Golgi apparatus to the plasma membrane, where it is processed by α- or β- and γ-secretase, following the non-amyloidogenic or amyloidogenic pathway, respectively [81,82,83,84].

α-secretase cleaves within the Aβ sequence of APP, thereby suppressing Aβ generation. Moreover, the resulting fragment of APP (APPsα) has been reported to have neurotrophic and neuroprotective properties [85]. APP molecules that fail to be cleaved by *α*-secretase can be internalized into endocytic compartments and, subsequently, the extracellular region of APP is processed by β-secretase (identified as BACE-1) to generate a β-carboxyl terminal fragment (β-CTF). Then, β-CTF is cleaved intramembraneously by a multimeric γ-secretase complex comprising four membrane proteins: presenilin (PS1 or PS2) as a catalytic subunit, nicastrin (NCT), anterior pharynx-1 (Aph-1), and presenilin enhancer 2 (Pen2) [86]. PS1 contains nine transmembrane domains (TMD1–9) and two catalytic aspartate residues in TMD6 and TMD7 [87]. During assembling of the complex, PS1 splits with the help of two different protease activities [88] into a loop between TMD6 and TMD7, creating N-terminal and C-terminal fragments. This γ-secretase-mediated cleavage produces Aβ peptides ranging from 38 to 43 amino acids [89]. In the brain, Aβ40 and Aβ42 peptides are the most frequently encountered and localized mainly extraneuronally [90]. The most neurotoxic and prone to aggregation are the long forms of Aβ (Aβ42 and Aβ43), in contrast to the short forms (Aβ38 and Aβ40) [91,92]. Elevated levels of Aβ42 with an increased ratio of Aβ42:Aβ40 leads to the generation of plaques and deposits, which accumulate toxic Aβ peptides [90]. The generation of these structures promotes microglia activation, oxidative stress, mitochondrial dysfunction, inflammatory responses, and synapse dysfunction and disturbs cellular communications, ultimately resulting in a cascade pathway to neuronal atrophy and AD [13,90,92]. Furthermore, a decrease in α-secretase (81% of normal) and a large increase in β-secretase (185%) are observed in the AD temporal cortex [93].

Cryo-electron microscopy (EM) of γ-secretase helped to understand the mechanism of its action. TMDs of each subunit are horseshoe shaped, with an extracellular domain located on a transmembrane helix [94]. PS1 has a hydrophilic, catalytic water-accessible pore [94]. According to cross-linking experiments, β-CTF initially interacts with the extracellular PS1 domain and then moves for proteolysis [95]. PS1 forms a hybrid β-sheet with a substrate at the catalytic pore, which extends the substrate’s spiral to deploy cleavage locations [96]. The location of binding sites of the γ-secretase inhibitor (GSI) and modulator (GSM) were also revealed [97]. GSI binds to the catalytic pore of PS1, while GSM to TMD1 of PS1. GSM inhibiting formation of long Aβ species was considered as a potential therapeutic agent for AD treatment [88].

To date, 19 FAD mutations have been identified in *APP*, and >150 in *PSEN1* have been reported [98]. These mutations were marked by increasing the relative production of Aβ42 [82,99]. Specifically, the transmembrane domains TMD of APP are found to be hotspots for FAD mutations, which are responsible for an increased fraction of relatively long Aβs [100,101,102]. Interestingly, the A673T mutation found in the Icelandic population and associated with a lower proportion of long forms of Aβs protects against AD [103]. However, the detailed mechanism is unclear.

Initially, APP processing studies dominated research in yeast models of Aβ. One of the advantages of using yeast models to study human APP processing is the possibility to express individual subunits of the human secretase complex to dissect their contribution, which is difficult to achieve in mammals due to the high abundance of these proteases. It was shown that yeast possesses an α-secretase-like activity and that most likely APP processing proceeds in the late Golgi complex. APP was expressed in a protease-deficient yeast strain and linked to the prepro-α-mating factor, which acts as a signal sequence. After Kex2 processing in the late Golgi complex and separation of the α-factor, a full-size APP is formed. It is further processed as evidenced by the detection of an N-terminal ectodomain in an external medium, while the resulting C-terminal fragment was the same size as the C-terminal fragment created by α-secretase cleavage in human cells. Blockage of transport from ER to the Golgi inhibited APP splitting [104]. Later, it was reported that two aspartyl proteases, encoded by YAP3 and MKC7, exhibiting α-secretase activity, were responsible for this cleavage [104,105]. A novel approach to identify APP processing secretases was based on engineering APP fragments containing the β-site, the transmembrane domain, and the C-terminal domain and fusion to yeast invertase with subsequent cleavage by human β-secretase [106]. The deletion of proteases Yap3 and Mkc7 inhibited yeast growth. Expression of human BACE-1, responsible for the β-secretase activity in humans, restored the growth in the absence of yeast proteases, which was indicative of the APP processing. This system, initially developed for the identification of novel secretases, was also applied for the screening of compounds that inhibit BACE-1 activity.

The precise role of the 99-aa membrane-bound C-terminal soluble fragment of APP (C99) remained poorly understood. To overcome this gap, an inducible C99-expressing yeast strain with compromised proteasomal activity was constructed, and formation of APP fragments possessing striking similarities to APP fragments present in AD patients was established [107]. Later, it was shown [108] that C99 acts as a lipid-sensing peptide, a mediator of cholesterol disturbances in AD, by inducing the internalization of extracellular cholesterol and its trafficking from the plasma membrane to the ER, potentially explaining early hallmarks of AD. Very recently [109], it was suggested that C99 fragments in yeast cells with compromised proteasomal activity can exert a myriad of cellular toxicities including protein aggregates, cellular stress and chaperone expression, electron-dense accumulations in the nuclear envelope/ER, abnormal DNA condensation, an induction of apoptosis, and others that are likely relevant to AD-associated neurodegeneration in humans.

Using the artificial substrate containing the C 1–55 (C55) fragment of APP linked to the Gal4 transcription factor, it was demonstrated that all four subunits of γ-secretase are essential for its protease activity [110]. Using the membrane-bound substrates (APP or Notch) linked with Gal4 and the Gal reporter, a system was established to monitor the cleavage reaction, evaluate the specificity of γ-secretase, find the release of different Aβ species (Aβ38, Aβ40, Aβ42, Aβ43, and Aβ45) in yeast microsomes, as well as to screen mutations and even to find chemicals that modify the activity of the protease, GSI, or GSM [82,110,111,112]. Notably, among different mutations created were those increasing the trimming of γ-secretase and reducing long Aβ species, especially Aβ42, in yeast and mammalian cells [111].

The yeast two-hybrid system is a useful molecular tool for studying protein–protein interactions. This tool is excellent for studying the interaction of proteins involved in the development of AD. With all its reliability, comparable to mass spectrometry, the yeast two-hybrid system has additional advantages such as low cost, time savings, and simplicity [113]. The yeast two-hybrid system has been utilized to detect proteins that interact with the extracellular [114] and the intracellular cytoplasmic domain of APP [115,116].

The yeast studies described above allowed better understanding of the processing of APP and, more importantly, resulted in a series of high-throughput models for the genetic screening of secretases and genetic/chemical screenings of modulators of Aβ42 aggregation and toxicity with potential therapeutic applicability.

#### 2.1.2. Yeast as a Model of Aβ Aggregation

Electron microscopy studies have determined that the extracellular senile plaques composed of a dense core of amyloid fibrils, aggregates of small peptides of 39–43 amino acids in length that have a cross-β structure [117,118], are associated with degenerating neurites, astrocytes, and astrocytic processes [119]. The intracellular neurofibrillary tangles are fibril aggregates of the hyperphosphorylated protein tau [120].

Generation of Aβ is localized in the plasma membrane. Thus, the formation of a toxic peptide occurs through a secretory pathway that leads to the transfer of Aβ to the trans-Golgi, endosomal compartments, and extracellular space [121].

Protein aggregation is enhanced at hydrophilic–hydrophobic interfaces, such as a cell membrane surface. The high concentration of the peptides accelerates the aggregation there. High concentrations of Aβ peptides initiate microglial infiltration, simultaneously triggering innate immune response against the aggregation. In addition, a β-hairpin structure of Aβ peptides (in Aβ40, for example), which is readily formed at the interface, accelerates the formation of an oligomer with the intermolecular β-sheet structure [122].

Recently, the small angle neutron scattering (SANS) technique was used to study aggregates and aggregation kinetics of Aβ [123,124]. Aβ40, Aβ42, and Aβp3–42 peptides were shown to have monomers with a radius of gyration of the order of 10 Å, while the oligomers and fibrils displayed differences in size and aggregation ability, with Aβp3–42 showing larger oligomers. These properties are strictly related to the toxicity of the corresponding amyloid peptide.

Yeast has been utilized to identify mechanisms regarding in vivo Aβ oligomerization, aggregation, toxicity, and the intracellular pathological properties of the peptide [13,121,125,126].

A yeast two-hybrid system (Aβ associated with the DNA-binding domain of LexA (bait) as well as a transactivation domain (prey)) was among the early promising model systems [127]. Aβ amino acids, on which its self-action and subsequent nucleation-dependent aggregation depend, were found, thus adding the argument that Aβ was capable of interaction with itself in vivo. Then, a yeast model system was elaborated based on fusion of Aβ42 to the C-terminus of the ER directing signal Kar2 (known as ssAβ42). This method allowed Aβ to enter the ER and target the secretory pathway, where it could undergo endocytosis and thus be transported through endocytic compartments potentially associated with AD. Aβ42 was toxic to yeast cells [128]. The ER is seriously impaired in AD neurons [129], and increased synthesis of Aβ peptides and their subsequent aggregation has been proposed to lead to increased levels of ER stress, having an effect on synaptic dysfunction in AD [129].

In connection with new data on the intracellular effects of Aβ, yeast model systems were created that made it possible to study the fusion of Aβ with prion proteins. Yeast prions are extensively studied [130]. They share some hallmarks with mammalian prion proteins or other amyloidogenic proteins found in the pathogenesis of AD. Chimeric constructs using mammalian and human aggregation-prone proteins or domains linked to fluorophores or to endogenous yeast proteins provide an opportunity for cytological or phenotypic discovering of AD-related protein in yeast cells, amyloid formation, propagation, aggregation, and/or toxicity [130]. Sup35 of *S. cerevisiae* is a known yeast prion protein, an essential translation termination factor, that can produce self-propagating amyloid aggregates, which results in a prion phenotype called [PSI+] [131,132]. The prion properties of Sup35p are related to its N-terminus. However, the N-terminus is not critical to the main function of the protein. The [PSI+] form cannot aggregate when the prion-forming domain is deleted, while replacing the prion-forming domain with Aβ restores the protein’s potential to aggregate [133].

Fusion of Aβ42 with MRF (Sup35p lacking N-terminal domain) and expression in yeast under the control of CUP1 (a copper-inducible promoter) caused aggregation into oligomers, as did the Sup35 protein with the entire amino acid sequence [134]. The possibility of interaction between Hsp104 and Aβ was also found. In addition to that, the oligomerization process is inhibited by the deletion of Hsp104 [134]. Furthermore, yeasts were also used to design novel screening systems for anti-prion compounds from chemical libraries [135].

Later [136], the previously demonstrated ability of the polymeric form of prion protein (PrP) and Aβ-GFP to form aggregates in yeast was confirmed [137,138,139,140]. It was shown that PrP- and Aβ-based aggregates produced in yeast exhibit certain amyloid properties. Moreover, colocalization between Aβ- and PrP-derived constructs fused to GFP in the yeast cell was demonstrated and was proven using FRET, confirming direct physical interaction between these proteins. Finally, co-expression in the yeast cell of Aβ and PrP(Sc) or its shortened derivatives fused to different fluorophores revealed PrP fragments involved in the interaction with Aβ: the 90–110 and 28–89 regions of PrP control the binding of proteinase-resistant PrP polymers to the Aβ peptide, with the 23–27 segment of PrP being dispensable for this interaction. These findings demonstrate the suitability of the yeast-based experimental assay for studying interactions between mammalian amyloidogenic proteins.

Following the pioneering work of Treush and colleagues [128], it was found that sending Aβ42 fused with GFP into the secretory pathway is necessary for the production of toxic species. Aβ42 fused to GFP retained the ability to aggregate, was toxic, and activated autophagy [126]. Moreover, yeast phosphatidylinositol binding clathrin assembly protein (PICALM) orthologs (Yap1802) were suggested to be associated with cellular toxicity, suggesting a similarity in the mechanisms of toxicity between mammals to yeast [141].

Finally, Nair and colleagues [142], by expressing Aβ-GFP in the complete *S. cerevisiae* genome-wide deletion mutant collection, identified mitochondrial dysfunction and impairments in transcriptional/translation regulation and especially in phospholipid metabolism as affecting Aβ42 aggregation.

The above data underline the value of yeast models in AD studies related to Aβ oligomerization and aggregation, as well as the resulting pathological cellular processes.

#### 2.1.3. Yeast as a Model for Toxicity of Aβ Aggregates

The large extracellular plaques are mainly inert and less toxic, although they might represent a reservoir for soluble Aβ. The most toxic types of Aβ42 in the brain are soluble oligomeric forms. In AD, they are associated with a positive correlation with neuronal death and cognitive deficits [143].

The first experiments on Aβ effects on cellular viability were performed by adding chemically synthesized Aβ to cells [121,144]. A rapid absorbance-based growth assay was created to test the toxicity of Aβ [143]. When freshly prepared, exogenous Aβ inhibited growth of yeasts (*Candida glabrata* and *S. cerevisiae*) [143,144]. However, varying methods of Aβ preparation and using yeast cells harvested at different growth phases led to conflicting results [145].

Even relatively recently, all attempts to obtain sufficient amounts of native Aβ42 in *S. cerevisiae* were unsuccessful, supposedly due to extremely rapid degradation or Aβ toxicity. Treush and colleagues were the first to do so (see above). In unboiled samples, significantly fewer Aβ40 oligomers have been revealed compared to Aβ42 oligomers using Western blotting. Importantly, an overexpression library of 5532 open reading frames (ORFs) was transformed into the Aβ42 screening strain with an intermediate Aβ toxicity, which greatly facilitated the detection of enhancers or suppressors of toxicity. A total of 17 enhancers and 23 suppressors of Aβ toxicity were found. Some of them were similar to human genes. Moreover, several of these genes had human homologues with AD risk factors, among them are the human homolog YAP1802, PICALM, and phosphatidylinositol binding clathrin, one of the most well-established risk factors for AD [146,147]. The experiments performed using transgenic *C. elegans* and primary rat cortical neurons supported results obtained from yeast, thus validating the model.

Recently, it was shown that the ER is significantly disrupted in AD neurons and suggested that elevated production of Aβ and subsequent aggregation enhances ER stress and consequently affects synaptic dysfunction in AD [129].

Yeast in which the prepro-α mating factor, Aβ42, and GFP were fused showed significant and reproducible cytotoxicity [141]. In addition, the ability of Aβ42 to act on mitochondria was discovered due to its ability to pass through intracellular membranes. The fusion protein containing the arctic Aβ42 mutant was found to be more toxic than the wild-type Aβ. Aβ without the prepro-α-mating factor signal sequence did not cause any clear cytotoxicity, further demonstrating that assembly of toxic species occurs under the influence of intracellular traffic pathways and PICALM yeast orthologues are associated with cellular toxicity [141]. However, these figures contradict others (see, for example, [47]), possibly because of differences in expression levels of involved constructs. Further studies will determine the role of PICALM and endocytosis and their involvement in Aβ aggregation occurring during different stages of AD [13,121].

In yeast constitutively producing native Aβ directed towards the secretory pathway, significant toxic effects were observed. These include lower growth rate, lower biomass yield, lower respiration rate, increased oxidative stress, symptoms of mitochondrial dysfunction, and ubiquitin proteasome dysfunction [126,148].

In a model constitutively expressing human native Aβ and having inducible promoters, authors [75] were able to show that the generation of Aβ monomers and oligomers in cells lowered growth rate biomass and respiratory rate related to increased production of ROS (increased oxidative stress) and decreased proteasomal activities.

Somewhat later, Chen’s group [149], utilizing bioreactor cultures (ensuring highly controlled conditions), found a synergistic effect between bioenergetics and stress in the ER that increases Aβ toxicity as well as inequality in Aβ40 and Aβ42 cytotoxicity. Significant disturbances in mitochondrial function (decreased respiration rate and ATP production), increased ROS levels leading to reduced growth rate and biomass yield, and impaired lipid synthesis were the result of Aβ42 expression, which caused sustained high ER stress and unfolded protein response (UPR), while Aβ40 expression induced only mild ER stress.

Consistent with previously reported data, Panaretou and Jones [150] showed that entering the secretory pathway, processing in, and output from the ER is essential to achieve the full Aβ42 toxic potential. Analysis of the yeast knockout collection showed a decrease in Aβ42 toxicity not only in strains damaged in ER–Golgi traffic and mitochondrial functioning but also in strains with damaged cell cycle and DNA replication stress response. By contrast, increased toxicity of Aβ42 was revealed in strains with injured organizations of the actin cytoskeleton, endocytosis, and the formation of multivesicular bodies. Since the latter has been proven to be necessary for the recovery of membrane damage in mammalian systems, they explored this issue in more detail in the yeast model and again confirmed the benefit of using yeast as a model in research of basic mechanisms of neurodegenerative disorders.

Using an improved yeast model [129] constitutively expressing Aβ42 and coupled with a synthetic genetic array (SGA) allowed researchers to observe an Aβ42-mediated reduction in respiratory function as well as elevated levels of ROS. More importantly, this approach allowed them to identify Aβ toxicity modifiers and investigate the interactions between genes and their related pathways. Using the gene enrichment analysis, it was determined that yeast mutants that implement the processes of protein secretion and degradation are more susceptible to Aβ toxicity. Defects in riboflavin metabolism and heat shock of the FMN1 mutant strain resulted in increased Aβ toxicity [129]. Thus, these studies have shown that the identification of Aβ toxicity modifiers and their associated mechanisms and the discovery of suitable therapeutic targets can be performed using yeast Aβ models used in conjunction with GWAS and/or SGA.

Using *S. cerevisiae* (with a multicopy plasmid containing the Aβ42 sequence) as an experimental model, it was found that Aβ alters the oxygen consumption and the activity of complex III and IV and increases ROS production into the mitochondria, where Cta1 and Sod2 play a crucial role in the regulation of the redox balance [148]. Accordingly, several mitochondrial dysfunctions, including elevated ROS levels, decreased membrane potential, and impaired oxygen consumption have been observed in mitochondria isolated from a strain expressing Aβ. Aβ inhibited Cym1, a presequence protease, thus causing the accumulation of preproteins; mitochondrial Aβ impaired the maturation of preproteins in mitochondria isolated from mouse and human brain tissue [151].

A yeast model system shows a toxic synergy between tyramine, a trace monoamine with sympathomimetic properties, and Aβ [152]. In yeast expressing native Aβ and treated with tyramine, a significantly higher level of ROS and native Aβ, especially on respiratory substrates, was observed. Tyramine may be considered as a contributing factor to the development of AD.

In addition to the growth defect, Aβ42 also activated the heat shock response [138] and, based on data from patients with AD, increased expression of heat shock protein is a protective response [153]. Indeed, by applying a combinatory genetic and proteomic approach to study intracellular Aβ42 toxicity in yeasts, the HSP40 family member Ydj1, the yeast orthologue of human DnaJA1, as a key factor in Aβ42-mediated cell death was revealed [154]. It was demonstrated that Ydj1/DnaJA1 physically interacts with Aβ42 (in yeast and mouse), stabilizing Aβ42 oligomers and mediating their translocation to mitochondria. Deletion of YDJ1 highly decreases co-purification of Aβ42 with mitochondria and protects against Aβ42-induced mitochondria-dependent cell death. Therefore, purified DnaJ chaperone delays Aβ42 fibrillization in vitro, and heterologous expression of human DnaJA1 increases generation of Aβ42 oligomers and their harmful translocation to mitochondria in vivo. Downregulation of the Ydj1 fly homologue, Droj2, increased stress resistance, mitochondrial morphology, and memory status in a *Drosophila melanogaster* AD model. With these data, an unexpected property of specific HSP40s to increase the toxicity of Aβ42 was discovered.

Thus, the cytotoxicity might occur due to inhibition of the proteasome [155], oxidative stress through ROS generation and defective mitochondria [156,157], alteration in endocytic efficiency [158], failure of Ca^2+^-signaling [159], and disturbance to synaptic receptor levels and activity [160]. This suggests that the ability to model intracellular Aβ toxicity and disruption of mitochondrial functions makes yeast-based Aβ models an extremely valuable tool for studying AD. In addition, in combination with GWAS studies, these models become an excellent platform for screening promising compounds that can modify Aβ cytotoxicity.

#### 2.1.4. Yeast as a Model of Aβ De-Aggregation

Over the past three decades, great efforts have been taken to try to diminish amyloid accumulation in the brain of Alzheimer’s patients, but to no avail. Symptomatic treating with inhibitors of acetylcholinesterase or N-methyl d-aspartate antagonists provided only limited protection [161]. The identification of new substances for preventing or alleviating Aβ toxicity was hampered by the lack of cell models suitable for high-throughput screens [47]. Yeasts are ideally suited for this purpose [143].

Compounds with anti-oligomeric effects were discovered during drug treatment of Duchenne muscular dystrophy [162]. This efficient and sensitive high-throughput screen was further improved [163] when Aβ was fused to the Sup35 lacking N-terminal domain. This approach identified 7 of the 1200 FDA-approved drugs that reduced Aβ oligomerization in yeast and reduced toxicity to PC12 cells and yeast expressing Aβ42 aggregates.

Additionally, the overexpression of the Yap1802 gene, the yeast ortholog of PICALM and the human AD risk factor, mildly recovered endocytosis in yeast cells and inhibited Aβ toxicity [128,141,163]. These findings were confirmed in *C. elegans* and rat hippocampal neurons in which PICALM expression partially averted Aβ-induced cell death [128], thus indicating preservation of Aβ mechanisms of toxicity. Later, it was found that both a fusion of the yeast Sup35 prion domain to a multimeric non-amyloidogenic protein and the expression of a mammalian amyloidogenic protein did not contribute to prion nucleation and its propagation [164].

Expression of Aβ fused to GFP in the folate-deficient yeast identified folate as a potential inhibitor of Aβ aggregation [165]. This model was also used to hallmark autophagy as a defense mechanism against Aβ toxicity. In the brain, autophagic vacuoles are significant reservoirs of intracellular Aβ, and in AD, the violation of the autophagy increases Aβ accumulation [166]. Simvastatin might also assist in preventing Aβ42 misfolding and aggregation in a dose-dependent manner irrespective of inhibition of ergosterol [167]. However, the exact cause of prevention remains to be identified.

More recently, an advanced assay system was developed based on yeast strains having a plasmid containing strong, constitutive glyceraldehyde-3-phosphate dehydrogenase or phosphoglycerate kinase promoters, making it possible to constitutively express Aβ fused to GFP at both terminuses of the peptide [126]. It was confirmed that GFP-Aβ42 is sequestered and selectively transported to the vacuole for degradation, and autophagy is the main pathway for aggregates cleavage [168]. Screening of 192 autophagy mutants detected *RRD1*, *SNF4*, *GCN4*, and *SSE1* genes involved in inhibiting of GFP-Aβ cleavage clearance [168]. By using the advanced yeast system [126], it was established that the majority of cells (young) in a growing population can readily and efficiently degrade the Aβ-GFP fusion protein, in contrast to older cells, which can be an important observation for AD therapeutic strategies.

The large-scale screening of substances potentially useful for saving or protecting cells from Aβ-induced cell death, using expressing GFP-Aβ wild-type cells and a mutant Atg8Δ with a deletion of Atg8, an autophagy marker, revealed autophagy enhancers. They include antihistaminic Latrepirdine (Dimebon™), enhancing autophagy by increasing yeast vacuolar activity and the transport of Atg8 to the vacuole, as well as rapamycin and SMER28. They protected yeast and mammalian cells from Aβ-induced cell death due to the ability to isolate aggregated GFP-Aβ42 into autophagic-like vesicles for subsequent degradation. Importantly, all these compounds decreased intracellular GFP-Aβ42 levels and Aβ toxicity [168,169]. Latrepirdine additionally facilitated better cognition and slowed down progression of AD in a mouse model [170].

A screen of phenotypes in yeasts expressing the Aβ peptide identified dihydropyrimidine-thiones (DHPM-thiones) that selectively saved from Aβ-induced toxicity by reducing Aβ levels and recovering vesicle trafficking [171]. Both single and co-treatments also reduced death of neurons expressing Aβ in a nematode, suggesting that DHPM-thiones target a common protective mechanism, being supposedly metal-dependent.

Expression of Aβ-GFP uses the complete *S. cerevisiae* genome-wide deletion mutant collection (~4600 mutants) to detect proteins and cellular processes influencing intracellular Aβ42 aggregation [142]. Mutants deficient in mitochondrial function, phospholipid metabolism, and transcriptional/translation regulation were found to affect Aβ42 aggregation. Specifically, Aβ42 was found to be very sensitive to altered phospholipid homeostasis.

Flavin mononucleotide (FMN) was detected to reduce Aβ toxicity [129]. According to SGA screening, FMN deficiency (in the FMN1 mutant) increased Aβ toxicity [129], and when the strain constitutively expressing Aβ was cultivated in media supplemented with FMN, reduced Aβ aggregate formation, decreased Aβ toxicity and improved cell viability were observed. Heat shock of the FMN1 mutant strain considerably increased misfolding and aggregation of the toxic peptide [129]. Hydrogen peroxide-mediated oxidative stress in the constantly Aβ-producing strain lowered cell viability. Supplementation with FMN increased cell viability in both Aβ-producing and control strains [129]. Overexpression of flavoprotein Dus2p showed better viability in DUS2 mutant expressing Aβ by reducing Aβ toxicity, indicating that DUS2 could take part in regulating Aβ toxicity, which makes it, along with FMN, a promising therapeutic agent for lowering Aβ toxicity.

Finally, small molecule screens detected several 8-hydroxyquinolines, including clioquinol and dihydropyrimidine-thiones, that were marked by their ability to synergistically, in a metal-dependent manner, mitigate Aβ toxicity via mechanisms including enhancing Aβ turnover, rescuing vesicle trafficking, and protecting against oxidative stress [47,121,171]. In *C. elegans* expressing Aβ, treatments by one or two compounds diminished death of neurons [171]. In transgenic mice models, treatment with clioquinol analogues lowered Aβ accumulation and dramatically improved learning and memory that was markedly deteriorated in the course of AD-like neuropathology [172]. Moreover, PBT2, the clioquinol analogue, showed a cognitive improvement in AD patients [173].

Obviously, yeast-based compound screenings to identify promising molecules that can mitigate Aβ pathology are extremely valuable. In several cases, the proposed mechanism of action of a compound, inferred from the results obtained from the yeast research, was confirmed on other, more complex AD models.

### 2.2. Yeast Models of Tau-Protein Aggregation and Toxicity

In recent years, the heterologous expression in yeast of genes encoding main biomarkers of neurodegenerative diseases has provided insight into many of the molecular mechanisms underlying AD. The value of yeast models in these studies is also due to the possibility of studying not only neuronal proteins but also their various post-translational modifications associated with the development of the disease. This property is extremely important for studying the pathological properties of the tau protein, which plays a key role in the development of a number of neurodegenerative diseases called tauopathies, which include AD. Unlike amyloid pathology, which, as noted above, develops well before the onset of AD symptoms, tau-related pathology coincides with clinical symptoms (cognitive decline and dementia) [174]. Decreased content of tau protein mitigates Aβ-mediated cytotoxicity, making tau an attractive therapeutic target [36]. During the AD process, tau becomes hyperphosphorylated, undergoing proteolytic cleavage with resulting modifications such as O-glycosylation, sumoylation, ubiquitinylation, acetylation, and others [76].

Tau protein is predominantly present in neuronal and glial axons, where it is of paramount importance for many physiological processes due to its influence on the dynamics of the microtubule system [175], regulation of axonal transport/lengthening/maturation, synaptic plasticity, and maintenance of DNA and RNA integrity [13]. Clearly, tau dysfunction can cause neurotoxicity by disrupting various processes in which it is involved.

The native tau protein has a disordered structure, having a tendency to take on a paperclip-like shape with closely located N- and C-terminal domains and repeating regions [13]. It is a charged and hydrophilic protein, which makes it highly soluble and stable in aqueous media over a wide range of pH and temperature. However, under pathological conditions, including AD, hyperphosphorylated (PP-tau) tau monomers dissociate from microtubules and tend to aggregate first into oligomers and then into neurofibrillary tangles, in which portions of their microtubule-binding domain, predominantly positively charged, are densely packed [176]. Tau protein function is regulated by several post-translational modifications, including phosphorylation, glycosylation, isomerization, acetylation, O-glycosylation, ubiquitination, deamidation, methylation, and oxidation [29,76]. During AD, tau distribution is changed, forming depositions in the somatodendritic compartment [177].

Tau phosphorylation has been well known. The protein contains 80 supposed serine/threonine phosphorylation sites and 5 potential tyrosine phosphorylation sites. Tau phosphorylation involves numerous kinases belonging to four different classes. Recently, GSK3α, GSK3β, MAPK13, and AMP-activated protein kinase were found to directly phosphorylate tau in various cell lines [178,179]. Tau dephosphorylation is a significant factor affecting its affinity for microtubules and, therefore, microtubule depolymerization. On the other hand, so-called hyperphosphorylation at certain epitopes (e.g., Thr181, Thr231, Ser202, Ser205, Ser214, Ser396, Ser404, Ser409, and Ser422) considerably changes the binding potential and stabilizing capacity properties of the tau protein, increasing the tendency for subsequent oligomerization and aggregation of tau into helical filaments and neurofibrillary tangles, which are characteristic of a group of neurodegenerative diseases called taupathies, including AD [13].

Not many studies have used yeast to study tau biology, in particular, its phosphorylation, conformation, and aggregation [180]. When overexpressed in *S. cerevisiae*, tau becomes hyperphosphorylated and acquires several pathological phosphoepitopes. Protein kinases Pho85 and Mds1, and yeast orthologues of human tau kinases, Cdk5 and GSK3β, respectively, have been shown to be important in regulating tau phosphorylation [13].

Yeast models have been useful for understanding tau modifications [76]. Tau isolated from the *S. cerevisiae* pho85∆ strain by anion exchange chromatography maintained its hyperphosphorylation [181]. The ability to purify these stable, pathologically relevant tau structures from *S. cerevisiae* cells created prerequisites for the use of the purified tau protein as an antigen source for mouse immunization [182]. This approach provides a considerable advantage over *E. coli*-based tau isolation and antibody production because tau does not undergo post-translational modification in bacterial cells. The antibody obtained by immunization of mice with 2N-/4R-tau isolated from the *S. cerevisiae* pho85∆ strain detects both mono- and oligomeric tau protein [182]. Now, this antibody is a component of the Digital Enzyme Immuno Assay (ELISA) platform, which allows to trace very low tau concentrations, thus revealing the potential of tau as a serum-based biomarker for AD.

As was inferred from the research using phosphospecific antibodies applied to the human tau protein isoforms (3R-tau and 4R-tau) expressed in *S. cerevisiae* [181], yeast kinases are able to recognize and phosphorylate pathological tau epitopes. Additionally, the MC-1 antibody, which binds to pathological tau filaments and their precursors, can also be used for tau detection [183,184]. Aggregated tau species are partially found in the sarcosyl-insoluble fraction (SinT-sarcosyl-insoluble tau) [181]. The yeast kinases Mds1 and Pho85, orthologues of the human kinases GSK-3β and CDK5, respectively, can directly phosphorylate tau.

At the same time, CDK5 has a regulatory function due to the ability to inhibit GSK-3β [185]. Deletion of Mds1 in *S. cerevisiae* resulted in a significant decrease in phosphorylation of epitopes AD2 (P-S396–P-S404), which is the target of human GSK-3β, and PG-5 (P-S409) [186]. The PG-5 site is a target for PKA but not for GSK-3β, which indicates the ability of Mds1 to indirectly regulate PG-5 phosphorylation [187,188]. Interestingly, deletion of Pho85 significantly increases the immunoreactivity of phosphospecific antibodies and the presence of tau in the sarcosyl-insoluble fraction [181]. Therefore, AD2 and PG-5 epitopes are associated with tau aggregation, while Pho85 kinase, such as mammalian CDK5 [189], indirectly downregulates phosphorylation and ultimately causes conformational changes and tau aggregation. Some additional characteristics of hyperphosphorylated tau were described in in vitro studies [181,190].

For a number of clinical FTDP-17 (frontotemporal dementia and Parkinsonism linked to chromosome 17) tau mutants, the main phosphorylation patterns and SinT levels were identified when expressed in wild-type, mds1Δ, and pho85Δ yeast strains [191]. Presumably, the effects of FTDP mutations are related to the ratio of 3R- and 4R-tau isoforms, which affects the ability of tau to bind to microtubules before the tau conformation changes [192,193]. The correlation of phosphorylation at the sites PG-5 and AD2 was found. Thus, phosphorylation of PG-5 can initiate subsequent phosphorylation of AD2 [191].

Experiments with the recombinant tau-4R and the mutant tau-P301L isolated from the wild-type yeast strain and Mds1 and Pho85 kinase mutants showed an inverse relationship between tau hyperphosphorylation and its ability to bind to microtubules [190]. Tau pathologically did not significantly affect the yeast growth rate.

In further studies, human tau and P301L and R406 mutants were expressed in wild-type yeast cells and Mds1 or Pho85 mutants. A decreased degree of phosphorylation of the PG-5 site (S409, Pho85 kinase target) was accompanied by a reduced degree of tau aggregation and enhanced tau binding to microtubules. The PG-5 site appears to play a key role in tau aggregation in yeast cells. Moreover, it was found that tau aggregation in yeast cells resulted in mitochondrial dysfunction and oxidative stress [191]. On the contrary, in human neuronal cells, oxidative stress caused tau dephosphorylation in the cdk5-dependent pathway [194]. This inconsistency necessitates a more detailed study of the oxidative stress as affected by tau properties. It became clear, however, that not only phosphorylation but also other mechanisms are involved in tau pathophysiology and that yeast models are useful in the search for and study of these possible mechanisms [174].

Using yeast models, an alternative mechanism for the regulation of tau phosphorylation, involving Ess1, the human orthologue of peptidyl prolyl cis/trans isomerase Pin1, was revealed [195]. A deficiency in Pin1 isomerase activity enhanced tau phosphorylation at the Thr231 site and reduced the growth rate of yeast, while tau expression enhanced cell death.

Expressing of tau and α-synuclein in yeast cells exerted a synergistic toxicity to cells [196,197]. An increase at the AD2 site of phosphorylation, and hence an elevated tau aggregation, has been found, again, with no decrease in the rate of yeast growth. Further research is also needed.

Another mechanism for regulating tau phosphorylation in yeast is inositol phosphate (IP) signaling, playing a central role in energy metabolism. A decline in the activity of inositol hexakisphosphate kinases Kcs1 and Vip1 (yeast orthologues of mammalian IP6 and IP7 kinases) enhances tau phosphorylation, most likely through the IP-mediated regulation of Pho85 kinase. Diminishing activity of Kcs1, Vip1, and Pho85 caused impairments in sphingolipid metabolism and reduction of sphingolipid biosynthesis, which, in turn, increases tau phosphorylation.

Humanized yeast models can be used to generate high affinity anti-tau monoclonal antibodies, for example, high-affinity monoclonal antibodies to human tau2N4R-ΔK280 were obtained [198]. Although most studies on tau protein were performed using the yeast *S. cerevisiae*, a traditional model organism, alternative yeast models have increasingly been proposed in recent years, and this coincides with the main stream in research on biomarkers of neurodegenerative diseases in yeast (see above).

Indeed, the research of phosphorylation, aggregation, microtubule binding, and cytotoxicity of the tau protein has benefited a lot from the use of yeast models. However, it remains unclear how these pathological effects of tau protein observed in yeast can recapitulate changes in human neurons during neurodegeneration. New research should shed light on this issue.

## 3. Concluding Remarks: The Need to Use Yeast for More In-Depth Study of AD

AD is a complex multifactorial disease and the most prevalent form of dementia, starting decades before the appearance of first cognitive symptoms [46]. Past studies of AD therapy provided valuable, albeit limited, insights. Currently, AD treatments are limited to only symptomatic management [10,199]. It is therefore essential to deepen our understanding of AD by detecting new risk factors and biomarkers and developing therapeutic strategies to treat or prevent the disease. Research reviewed above clearly indicates that yeast models, together with other more simple eukaryotic models including animal models, *C. elegans* and *Drosophila* (reviewed in [70]), significantly contributed to understanding Aβ and tau biology. These models combined with an enhanced suite of technologies allowed for high-throughput screening of drugs affecting Aβ oligomerization, aggregation and toxicity, as well as tau hyperphosphorylation.

The effects of a number of compounds were first established in yeast model systems. The most notable examples include latrepirdine, which enhances autophagy by increasing the transport of the autophagy marker Atg8 into the vacuole and, consequently, reduces levels of Aβ42 aggregation [169]. Clioquinol [47] and dihydropyrimidine-thiones [171] reduced Aβ42 toxicity in yeast via a metal-dependent mechanism of action. In addition, in yeasts with overexpressed Yap1802, a yeast orthologue of PICALM, seven well-known compounds (bromperidol, minocycline, pramoxine, dyclonine, haloperidol, azaperone, and tamoxifen citrate) were able to reduce Aβ42 oligomerization and cellular toxicity [163]. At the same time, the beneficial action of many of the above compounds was confirmed in other model systems, including mammalian, thus reinforcing the utility of yeast models.

A great deal of research has been devoted to attempts to prevent the formation of Aβ aggregates or modulating and reducing Aβ toxicity. However, as noted above, there is no correlation between amyloid pathology and clinical manifestations such as cognitive decline [200]. Moreover, although amyloid pathology is developing well before AD symptom onset, it is not the earliest characteristic of AD. Neurons, as well as post-mitotic and excitable cells, have high energy needs accommodated almost exclusively by the mitochondrial oxidative phosphorylation system to meet their energy demands. Mitochondria play a crucial role in maintaining synaptic function. Energy deficiency and high levels of mitochondrial division (fragmentation) are important early events promoting synaptic deficiency and neural cell death in AD [201,202].

Future yeast models should apply not only *S. cerevisiae* but also so-called non-convential yeasts. For some of them, due to their prominent industrial potential to utilize waste substrates for eco-friendly production of green chemicals, for a short time there was developed a genetic toolbox, ranging from protein expression to iterative gene integration and Golden-gate cloning, gene/genome editing systems, such as CRISPR-Cas9, genomic modifications of transposons, and TALENs-based genome editing technologies [72].

New yeast models should be focused on the developing of new high-throughput systems for the analysis and disclosure of AD mechanisms, as well as on identification of affordable non-invasive biomarkers for the earliest symptoms of AD, the earliest violations in various cellular networks, as well as at “ohmic” levels (Figure 2). For improving classification of risk factors, early diagnosis and timely treatment starting should be at the center of attention for AD researchers [46]. However, it should be noted that yeast, as a unicellular organism, fails as a model to study the multicellularity and cell–cell interactions, which are particularly important in the neuronal cross-talk that is of major importance to neurodegeneration. Yeasts lack neuron-specific morphological structures, such as dendrites, axons, and synapses. Consequently, the underlying neuron-specific molecular inventories are missing. Therefore, disease-associated processes uncovered in yeasts must be validated in neuronal model systems and eventually in human studies [46].

## Figures and Tables

**Figure 1 ijms-24-09791-f001:**
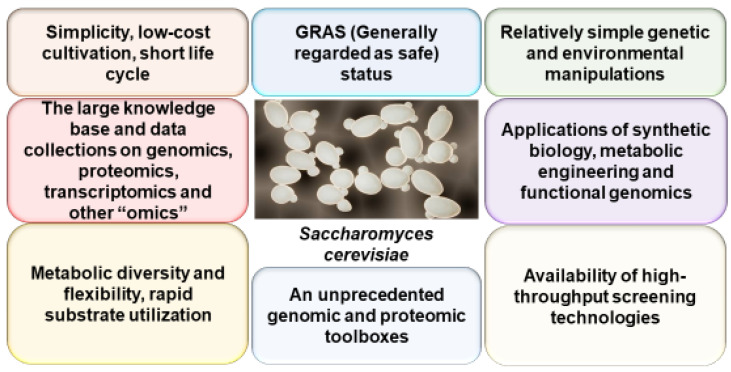
Advantages of yeasts over other AD models. The picture of *Saccharomyces cerevisiae* was taken from Wikipedia.

**Figure 2 ijms-24-09791-f002:**
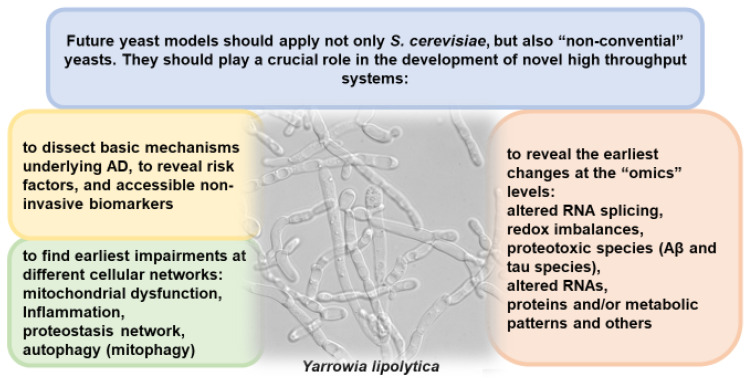
Future application of yeast models. The picture of *Yarrowia lipolytica* was taken from Wikipedia.

## Data Availability

No new data were created or analyzed in this study. Data sharing is not applicable to this article.

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
