# Peer review of "Alzheimer’s Disease: Significant Benefit from the Yeast-Based Models"

_ijms, 2023, doi:10.3390/ijms24129791_

Round 1

Reviewer 1 Report

Authors presented a comprehensive review for implication of the yeast model in the investigation of Alzheimer’s Disease. The manuscript is well written and covers the most recent article on the topic. I suppose that this article will be interesting for different scientists and can be published after minor revision.

Minor issues 

According to the Uniprot Database genes coding presenilins are called PSEN1 and PSEN2, and short names of the proteins — PS-1 and PS-2. Slightly different designations can be found in the text. Also, I encourage authors to recheck the usage of abbreviations of genes and proteins across the text and use italic when appropriate. For instance, on line 244 APP and PSEN1 correspond to the gene names and should be italicized; FMN1 on line 594. 

I would suggest to use abbreviation AD in only one sense, but the two variants are used now (lines 47 and 55)

On lIne 60 the word “century” seems to be missing.

On line 197 “(Ab)” should be replaced by “Ab”.

I would suggest to describe in more detail the usage of the two-hybrid system in the investigation of AD in yeasts. The corresponding paragraph comprises only one sentence. I think that several examples could be listed here. Another short paragraph is presented on lines 580-581.

I would suggest to mention in the manuscript another important paper revealing interaction region between Ab and PrP: Rubel, A. a, Ryzhova, T. a, Antonets, K. S., Chernoff, Y. O., & Galkin, A. (2013). Identification of PrP sequences essential for the interaction between the PrP polymers and Aβ peptide in a yeast-based assay. Prion, 7(6), 1–8. https://doi.org/10.4161/pri.26867

Description of the PICALM abbreviation is given only on line 538, however, the term is used for the first time on line 361.

I would suggest adding an illustration or summary table in the conclusion to summarize the advantages or disadvantages of the yeast model in investigations of different aspects of AD.

Author Response

REVIEWER 1 

First of all, we would like to thank the referee for useful comments and advises and are ready to answer the questions and criticism raised.

Authors presented a comprehensive review for implication of the yeast model in the investigation of Alzheimer’s Disease. The manuscript is well written and covers the most recent article on the topic. I suppose that this article will be interesting for different scientists and can be published after minor revision.

Minor issues

According to the Uniprot Database genes coding presenilins are called PSEN1 and PSEN2, and short names of the proteins — PS-1 and PS-2. Slightly different designations can be found in the text. Also, I encourage authors to recheck the usage of abbreviations of genes and proteins across the text and use italic when appropriate. For instance, on line 244 APP and PSEN1 correspond to the gene names and should be italicized; FMN1 on line 594.

Response: Done.

I would suggest to use abbreviation AD in only one sense, but the two variants are used now (lines 47 and 55)

Response: Done.

On line 60 the word “century” seems to be missing.

Response: Done.

On line 197 “(Ab)” should be replaced by “Ab”.

Response: Done.

I would suggest to describe in more detail the usage of the two-hybrid system in the investigation of AD in yeasts. The corresponding paragraph comprises only one sentence. I think that several examples could be listed here. Another short paragraph is presented on lines 580-581.

Response: We added a brief description of yeast two-hybrid systems, however, most of the examples of the use of this system are not directly related to the AD research. The most relevant examples are given in the paragraph. The short paragraph presented on lines 580-581 is deleted.

I would suggest to mention in the manuscript another important paper revealing interaction region between Ab and PrP: Rubel, A. a, Ryzhova, T. a, Antonets, K. S., Chernoff, Y. O., & Galkin, A. (2013). Identification of PrP sequences essential for the interaction between the PrP polymers and Aβ peptide in a yeast-based assay. Prion, 7(6), 1–8. https://doi.org/10.4161/pri.26867

Response: Thank you very much. We added a paragraph with three additional references.

Description of the PICALM abbreviation is given only on line 538, however, the term is used for the first time on line 361.

Response: Done.

I would suggest adding an illustration or summary table in the conclusion to summarize the advantages or disadvantages of the yeast model in investigations of different aspects of AD.

Response: We added two illustrations. The first describes the advantages of yeasts over other AD models. The second is devoted to possible future applications of yeast models in the AD research.

Reviewer 2 Report

This article is devoted to an interesting and current topic related to the use of yeast model systems to identify factors involved in Alzheimer's disease. The authors describe the experimental data on this topic in sufficient detail. Although the physiology of Saccharomyces cerevisiae and brain neurons differs significantly, the use of yeast models does provide certain advantages for the analysis of some aspects of neurodegenerative diseases. In general, the Review can be useful for researchers who are working on this topic. However, some fragments of the text are not directly related to the topic of the special issue. There are other comments and additions to the text of the manuscript.

1.      In the Introduction, the authors discuss the etiology of Alzheimer's disease and factors contributing to the aggregation of Aβ and tau. It is worth adding that Aβ aggregation is co-facilitated by its isomerization. It is shown that injection of the isoAsp7-containing Aβ(1-42), an abundant age-dependent Aβ isoform, significantly accelerates Abeta aggregation in the brains of transgenic mice  (https://pubmed.ncbi.nlm.nih.gov/23670398/).

2.      The section 2.1.1 titled -  Yeast as a model of APP processing. However, a large fragment  of  the text (Lines 197 – 250) is devoted to describing APP processing without regard to research in the yeast model. In my opinion, this description should be moved to the Introduction section and greatly shortened.

3.      The section 2.1.2. Yeast as a model of Aβ aggregation.

The data related to interaction of Aβ-GFP with mammalian Prion Protein aggregates should be considered in the Review  (https://www.tandfonline.com/doi/full/10.4161/pri.26867).

4.      Line 352. Sup25p must be replaced with Sup35p. The SUP35 gene encodes the prionogenic protein Sup35, which corresponds to the prion factor [PSI+].

5.      2.1.3. Yeast as a model for toxicity of Aβ aggregates

Describing the toxicity of extracellular Aβ to yeast cells, it should be mentioned that the mechanism of toxicity is different from the one that causes neuronal death. In the brain, Aβ oligomers bind to specific neuronal receptors. These receptors are absent in yeast cells.

6.      Line 474  Poor sentence wording: “Growth on glucose showed minimal growth inhibition”

7.      Line 513   Aβ-associated cell toxicit must be replaced with Aβ-associated cell toxicity.

8.      Lines 524 – 528. The biological significance of the described experiment is unclear.

9.      Line 530. The authors write: “This efficient and sensitive high-throughput screen was further improved [157], when Aβ was fused to the Sup35”.

In fact, the Aβ sequence was fused to the sequence encoding the C domain of the Sup35 protein. Fusion with the full-length Sup35 protein does not make sense.

10.  Lines 609 – 619. The authors describe experiments in a yeast system where the effect of certain chemical compounds on Aβ aggregation was shown. All this is true, but the effect of these substances was proved much earlier in other model systems. This is worth mentioning.

11.  Line 696.  Replace " and tau" with "of tau"

12.  Lines 762 – 765. The authors write that the number of articles dedicated to the tau protein in the yeast system is small. In my opinion, this statement is redundant.

13.   Lines 887 – 921 Concluding remarks.  

This part of the Concluding remarks section needs to be completely rewritten. The authors discuss the role of mitochondria in the regulation of neurodegenerative diseases. There are 18 references, and only one of them is related to yeast model systems. The final section should be rewritten in accordance with the title of the article and the subject of the special issue. I would advise to focus on those chemical compounds and proteins, the effects of which were first established in yeast model systems. It also makes sense to discuss further prospects for the use of yeast to search for factors involved in Alzheimer's disease.

Author Response

REVIEWER 2

First of all, we would like to thank the referee for useful comments and advises and are ready to answer the questions and criticism raised.

This article is devoted to an interesting and current topic related to the use of yeast model systems to identify factors involved in Alzheimer's disease. The authors describe the experimental data on this topic in sufficient detail. Although the physiology of Saccharomyces cerevisiae and brain neurons differs significantly, the use of yeast models does provide certain advantages for the analysis of some aspects of neurodegenerative diseases. In general, the Review can be useful for researchers who are working on this topic. However, some fragments of the text are not directly related to the topic of the special issue. There are other comments and additions to the text of the manuscript.

  1. In the Introduction, the authors discuss the etiology of Alzheimer's disease and factors contributing to the aggregation of Aβ and tau. It is worth adding that Aβ aggregation is co-facilitated by its isomerization. It is shown that injection of the isoAsp7-containing Aβ(1-42), an abundant age-dependent Aβ isoform, significantly accelerates Abeta aggregation in the brains of transgenic mice (https://pubmed.ncbi.nlm.nih.gov/23670398/).

Response: Done.

  1. The section 2.1.1 titled - Yeast as a model of APP processing. However, a large fragment of  the text (Lines 197 – 250) is devoted to describing APP processing without regard to research in the yeast model. In my opinion, this description should be moved to the Introduction section and greatly shortened.

Response: So far, all yeast model reviews have followed the traditional pattern of providing the material - a rather detailed overview of the disease, followed by a list and analysis of works on individual topics in special sections. We decided to move away from this stereotype by briefly outlining the general (largely known, moving from one review to another) information about AD, and by describing in more detail the extent of the study of specific processes underlying the disease. It seems to us that this way of presenting the material provides an opportunity to better assess what has been done on yeast and the contribution of yeast models to the general understanding of particular processes involved in AD.

  1. The section 2.1.2. Yeast as a model of Aβ aggregation.

The data related to interaction of Aβ-GFP with mammalian Prion Protein aggregates should be considered in the Review 

Response: Thank you very much. We added a paragraph with three additional references.

  1. Line 352. Sup25p must be replaced with Sup35p. The SUP35 gene encodes the prionogenic protein Sup35, which corresponds to the prion factor [PSI+].

Response: Done.

  1. 2.1.3. Yeast as a model for toxicity of Aβ aggregates

Describing the toxicity of extracellular Aβ to yeast cells, it should be mentioned that the mechanism of toxicity is different from the one that causes neuronal death. In the brain, Aβ oligomers bind to specific neuronal receptors. These receptors are absent in yeast cells.

Response: You are absolutely right. Obviously, the mechanism of toxicity must be different. Yeasts are devoid of both neuronal and leukocyte receptors of Aβ. But there are other mechanisms in yeast that can cause toxicity including well documented induction of mitochondrial dysfunction and oxidative stress, disrupting the dynamics of mitochondria (excessive mitochondrial fragmentation) and other implications of energy shortage.

  1. Line 474 Poor sentence wording: “Growth on glucose showed minimal growth inhibition”

Response: The phrase is changed to “Cultivation on glucose showed minimal growth inhibition”

  1. Line 513 Aβ-associated cell toxicit must be replaced with Aβ-associated cell toxicity.

Response: Done.

  1. Lines 524 – 528. The biological significance of the described experiment is unclear.

Response: Two sentences were removed.

  1. Line 530. The authors write: “This efficient and sensitive high-throughput screen was further improved [157], when Aβ was fused to the Sup35”.

In fact, the Aβ sequence was fused to the sequence encoding the C domain of the Sup35 protein. Fusion with the full-length Sup35 protein does not make sense.

Response: The phrase was changed to “This efficient and sensitive high-throughput screen was further improved [157], when Aβ was fused to the Sup35 lacking N-terminal domain”

  1. Lines 609 – 619. The authors describe experiments in a yeast system where the effect of certain chemical compounds on Aβ aggregation was shown. All this is true, but the effect of these substances was proved much earlier in other model systems. This is worth mentioning.

Response: We did. “In transgenic mice models, treatment with clioquinol (analogue) compounds inhibited Aβ accumulation and resulted in a dramatic improvement in learning and memory, accompanied by marked inhibition of AD-like neuropathology [166]. Moreover, treating AD patients with PBT2, the clioquinol analogue, indicated a cognitive improvement in AD patients [167]”.

  1. Line 696. Replace "and tau" with "of tau"

Response: The phrase was changed to “As was inferred from the research using phosphospecific antibodies applied to the human tau protein isoforms (3R-tau and 4R-tau) expressed in S. cerevisiae [175], yeast kinases are able to recognize and phosphorylate pathological tau epitopes.”

  1. Lines 762 – 765. The authors write that the number of articles dedicated to the tau protein in the yeast system is small. In my opinion, this statement is redundant.

Response: The sentence was removed.

  1. Lines 887 – 921 Concluding remarks.

This part of the needs to be completely rewritten. The authors discuss the role of mitochondria in the regulation of neurodegenerative diseases. There are 18 references, and only one of them is related to yeast model systems. The final section should be rewritten in accordance with the title of the article and the subject of the special issue. I would advise to focus on those chemical compounds and proteins, the effects of which were first established in yeast model systems. It also makes sense to discuss further prospects for the use of yeast to search for factors involved in Alzheimer's disease.

Response: The Concluding remarks section was completely rewritten. It was significantly reduced, as well as the number of references, became more compact and was strictly consistent with the title of the manuscript. On the other hand, we have added some new references and, as recommended, listed those substances and proteins that were first discovered in yeast. We also stressed that future yeast models should apply not only S. cerevisiae, but also so-called non-convential yeasts and be focused on revealing new factors involved in AD, as well as accessible non-invasive biomarkers of the earliest manifestations of AD.

Reviewer 3 Report

This is an interesting review potentially worth of publication. However, some improvements are required first.

The abstract appears too lengthy. A significant part of it may be more suitable in the Introduction or text. In any case, please check that the abstract fits the formal limits for its size.

In some places the authors simply list the publications not trying to critically analyze them. For example, the yeast Abeta aggregation model (Section 2.1.4.) is rather questionable. At least some of the cited publications use Abeta fusions to GFP or Sup35. Then, they observe small oligomers, mainly dimers. Are they really amyloids? We know that Abeta starts aggregating only when it is proteolytically processed at both ends. So, when adding GFP or Sup35, we reverse the situation back and restrict the ability of Abeta to polymerize. The added proteins may sterically interfere with Abeta amyloid formation. The described Abeta oligomers are much smaller than yeast prions, which contain from ten to several tens of monomers, and this is likely due to inability of Abeta amyloid to accommodate more than two-three globular subunits around Abeta amyloid core. (A longer linker between the two parts was required.) An amyloid structure of two monomers could exist, but it is unlikely to propagate in yeast, since propagation requires occasional division of significantly larger oligomers, which were not shown. And if it is not an amyloid, then could it serve as a model for the disease process? I do not insist, but this is my impression of the works cited in this chapter.

The paper is quite long and could benefit from shortening by omitting the least interesting citations. One candidate is the paragraph at lines 517-527, which has little relation to the section topic, de-aggregation.

The conclusion is also too long.

The language is generally good, but still requires improvements, and some points I note below.

Line 16: mechanistic?

Ln 58: 70-80% - I misunderstood at first that the remaining cases are dementia of different, non-AD origin.

Ln 60: better "MOST devastating diseases humanity faced in the 21st CENTURY"

Ln64: is the shift to Ab42 the cause or the consequence of AD?

Ln65: amyloidogenic, (+ comma)

Ln66: add or remove a comma

Ln 69: tend to aggregate first into oligomers and then into…

Ln71: - comma

Ln84-91: A too long sentence

Ln91: was?

Ln113: urgent?

Ln154: lipolytucsa

Ln158: remembering => resembling or reminding

194: Ref 77 is only about Huntington's disease in yeast

281: exert

280-284: check sentence

289: monitor

334: was toxicity => is toxic

336: abnormal(?) levels of ER stress – correct?

346: factor, that has a strong tendency to => factor that can

350: was deleted, while replacing of the => is deleted, while replacing the

352: with MRF (Sup25p: MRF was not defined, and apparently Sup35p

376: fiRst

380: state => phase

446: mediatrd educed => mediated reduction in

447: enabled => allowed

457-8: check language

472: a couple of words are needed about what is tyramine, its role in the Abeta story

513: toxicit has been hindered by a lack => toxicity has been hindered by the lack

514-516: rephrase the sentence

517: in description of this work, it is unclear, what is the aggregation state of the fresh Aβ42, what causes toxicity? Monomers? Non-amyloid oligomers? Amyloid oligomers? I would think that only the latter can model the disease.

525: I have never heard about "NaOH toxicity". Apparently, high pH is detrimental, but "toxic" is not used. Also: how this work helps to understand the disease?

577: reduce

802: Pin1 and Ess1 are the same protein, according to the yeastgenome.org, and the lack of it is lethal, not just detrimental.

891: harnfull

The language is generally good, but still requires improvements. Some points I noted, but this is not all, and some places may need rephrasing. 

Author Response

REVIEWER 3

First of all, we would like to thank the referee for useful comments and advises and are ready to answer the questions and criticism raised.

This is an interesting review potentially worth of publication. However, some improvements are required first.

The abstract appears too lengthy. A significant part of it may be more suitable in the Introduction or text. In any case, please check that the abstract fits the formal limits for its size.

Response: The abstract was considerably reduced, almost to formal limits.

In some places the authors simply list the publications not trying to critically analyze them. For example, the yeast Abeta aggregation model (Section 2.1.4.) is rather questionable. At least some of the cited publications use Abeta fusions to GFP or Sup35. Then, they observe small oligomers, mainly dimers. Are they really amyloids? We know that Abeta starts aggregating only when it is proteolytically processed at both ends. So, when adding GFP or Sup35, we reverse the situation back and restrict the ability of Abeta to polymerize. The added proteins may sterically interfere with Abeta amyloid formation. The described Abeta oligomers are much smaller than yeast prions, which contain from ten to several tens of monomers, and this is likely due to inability of Abeta amyloid to accommodate more than two-three globular subunits around Abeta amyloid core. (A longer linker between the two parts was required.) An amyloid structure of two monomers could exist, but it is unlikely to propagate in yeast, since propagation requires occasional division of significantly larger oligomers, which were not shown. And if it is not an amyloid, then could it serve as a model for the disease process? I do not insist, but this is my impression of the works cited in this chapter.

Response: ”It was shown that PrP- and Aβ-based aggregates, produced in yeast exhibit certain amyloid properties” (Aleksandr A Rubel  , Tatyana A Ryzhova, Kirill S Antonets, Yury O Chernoff, Alexey Galkin. Identification of PrP sequences essential for the interaction between the PrP polymers and Aβ peptide in a yeast-based assay. Prion 2013;7(6):469-76. doi: 10.4161/pri.26867).

The paper is quite long and could benefit from shortening by omitting the least interesting citations. One candidate is the paragraph at lines 517-527, which has little relation to the section topic, de-aggregation.

Response: Due to the addition of two figures and, as recommended, some clarifications, generalizations, etc., the article became longer by 1 page.

The conclusion is also too long.

Response: The Concluding remarks section was completely rewritten. It was significantly reduced, as well as the number of references, became more compact and was strictly consistent with the title of the manuscript.

The language is generally good, but still requires improvements, and some points I note below.

Response: We appreciate all your suggestions.

Line 16: mechanistic?

Response: Was deleted

Ln 58: 70-80% - I misunderstood at first that the remaining cases are dementia of different, non-AD origin.

Response: Likely yes, because predominant doesn’t mean the only possible

Ln 60: better "MOST devastating diseases humanity faced in the 21st CENTURY"

Response: The phrase is changed to “AD is recognized by the World Health Organization as a global public health priority and as one of the most devastating diseases humanity faced in the 21st century [14].”

Ln64: is the shift to Ab42 the cause or the consequence of AD?

Response: The phrase was replaced by ”A rebalancing between Aβ40 and Aβ42 in favor of the longer form, which is highly amyloidogenic hydrophobic, more prone to aggregation and toxic, is a prerequisite for the development of AD”.

Ln65: amyloidogenic, (+ comma)

Response: Done.

Ln66: add or remove a comma

Response: The comma was added.

Ln 69: tend to aggregate first into oligomers and then into…

Response: The phrase is changed to “tend to aggregate first into oligomers and then in neurofibrillary tangles”

Ln71: - comma

Response: Done.

Ln84-91: A too long sentence

Response: It was reduced.

Ln91: was?

Response: Rather yes, in the text there is an explanation why. However, some recent AD related studies continue to support the key role of Aß (Sorrentino V et al. Enhancing mitochondrial proteostasis reduces amyloid-beta proteotoxicity. Nature 552, 187–193 (2017); Zhu B et al. ER-associated degradation regulates Alzheimer’s amyloid pathology and memory function by modulating gamma-secretase activity. Nat Commun 8, 1472 (2017)).  To deny the toxicity of Aß aggregates is meaningless, the question is whether they are causative factors to induce oxidative stress and mitochondrial dysfunction or vice versa, oxidative stress and mitochondrial dysfunction will initiate aggregate formation.

Ln113: urgent?

Response: Is changed to “The need”

Ln154: lipolytucsa

Response: Is changed to “lipolytica

Ln158: remembering => resembling or reminding

Response: Is changed to “reminding”

194: Ref 77 is only about Huntington's disease in yeast

Response: Thank you very much. This reference was removed.

281: exerted

Response: Changed to “exert”

280-284: check sentence

Response: The sentence was removed.

289: monitor

Response: Changed to “monitor”

334: was toxicity => is toxic

Response: Changed to “was toxic”

336: abnormal(?) levels of ER stress – correct?

Response: Changed to “increased levels of ER stress”

346: factor, that has a strong tendency to => factor that can

Response: Changed to “factor that can”

350: was deleted, while replacing of the => is deleted, while replacing the

Response: Done.

352: with MRF (Sup25p: MRF was not defined, and apparently Sup35p

Response: Is replaced by “Sup35p”

376: first

Response: Done.

380: state => phase

Response: Is replaced  by “phase”

446: mediatrd educed => mediated reduction in

Response: Is replaced by “mediated reduction in”

447: enabled => allowed

Response: Is replaced by “allowed”

457-8: check language

472: a couple of words are needed about what is tyramine, its role in the Abeta story

Response:  We added a few words about tyramine, remaining paragraph part was removed.

Tyramine is a trace monoamine with sympathomimetic properties. May be considered as a risk or a contributing factor to the development of AD.

513: toxicit has been hindered by a lack => toxicity has been hindered by the lack

Response: Done.

514-516: rephrase the sentence

Response: The sentence was removed.

517: in description of this work, it is unclear, what is the aggregation state of the fresh Aβ42, what causes toxicity? Monomers? Non-amyloid oligomers? Amyloid oligomers? I would think that only the latter can model the disease.

Response: “Here we describe a method to produce Abeta in oligomeric form and the comparison of stable fibrillar and non-fibrillar forms in cell toxicity studies in water, achieved through the use of yeast”. Oligomeric form of Aβ42 was used

525: I have never heard about "NaOH toxicity". Apparently, high pH is detrimental, but "toxic" is not used. Also: how this work helps to understand the disease?

Response: The paragraph was removed.

577: reduce

Response: Done.

802: Pin1 and Ess1 are the same protein, according to the yeastgenome.org, and the lack of it is lethal, not just detrimental.

Response: Done.

891: harnfull

Response: Is replaced  by “harmful”

Reviewer 4 Report

This is a fine review that is useful, albeit there are others that are similar in the literature (even some published in IJMS).

line 45: APP should be amyloid precursor protein, not Ab

There are several statements in the paper that go too far, for example, "Research reviewed above clearly indicates that 693 the greatest progress came in understanding Aβ and tau biology from yeast models". The authors should better recognize the contributions of other assays and organisms. For example, how many drugs identified in yeast are disease modifying therapeutics? Though the review is focused on yeast, other systems and methods should be recognized.

The extent to which the yeast aggregated form match those from human brain (eg the important cryo-EM studies of Scheres) is likely low. This should be critically discussed with balance, rather than trying to sell the reader on the ubiquitous importance of yeast. Without doubt, yeast are an important model, but the review seems biased towards yeast over all else. I recommend the authors to balance their work in the large amyloid field to recognize the utility of yeast models without overselling these methods, which have serious and not fully discussed limitations (the key one being the degree to which they recapitulate human pathology). 

Author Response

Reviewer 4

First of all, we would like to thank the referee for useful comments and advises and are ready to answer the questions and criticism raised.

“This is a fine review that is useful, albeit there are others that are similar in the literature (even some published in IJMS).”

Response: “Several excellent reviews have recently dealt with specific aspects of AD, using yeast model [13,61,74-77]” (lines 172-173). Among them are the reviews you mention: Dhakal S, Macreadie I. Protein Homeostasis Networks and the Use of Yeast to Guide Interventions in Alzheimer's Disease. Int J Mol Sci. 2020; Seynnaeve D et al., Recent Insights on Alzheimer's Disease Originating from Yeast Models. Int J Mol Sci. 2018 Jul 3;19(7):1947.

They are excellent, often cited. We also cite them repeatedly in our review.

We added “However, lack of effective AD therapy, despite tremendous efforts in elucidating the molecular and cellular players involved in AD pathology, the resisted idea of dementia and AD as a fate (Andersson MJ, Stone J. J, Best Medicine for Dementia: The Life-Long Defense of the Brain.  Alzheimers Dis. 2023 May 20. doi: 10.3233/JAD-230429), the growing hope of researchers and the disappointed public in finding effective means to combat AD by using different approaches, including models, among them yeast-based, has motivated us to write a new review to sum up that has been new since the last AD reviews, and share our thoughts on how best to mitigate or to slowdown AD progression” (lines 173-179).

“line 45: APP should be amyloid precursor protein, not Ab”

Response: done.

“There are several statements in the paper that go too far, for example, "Research reviewed above clearly indicates that the greatest progress came in understanding Aβ and tau biology from yeast models". The authors should better recognize the contributions of other assays and organisms. For example, how many drugs identified in yeast are disease modifying therapeutics? Though the review is focused on yeast, other systems and methods should be recognized.

The extent to which the yeast aggregated form match those from human brain (eg the important cryo-EM studies of Scheres) is likely low. This should be critically discussed with balance, rather than trying to sell the reader on the ubiquitous importance of yeast. Without doubt, yeast are an important model, but the review seems biased towards yeast over all else.”

Response: After reading the review again, I (RAZ) realized that you are absolutely right. My passion for yeast is due in part to personal motives. The thing is, I’ve been doing yeast my whole life. I am the person who, long ago, first isolated yeast mitochondria, not mitochondria fragments but mitochondria, showed the features of energy of yeast mitochondria, etc. For these works I was awarded with our prestigious national prize, for the best work in the field of biochemistry. Foreign colleagues who have difficulty pronouncing my surname have applied to the management of our institute to find “madam yeast science”. Secondly, the review is not dedicated to all models, namely yeast.

Of course, according to your criticism, we made the appropriate adjustments.

The phrase “Research reviewed above clearly indicates that the greatest progress came in understanding Aβ and tau biology from yeast models” was replaced by “Research reviewed above clearly indicates that yeast models, together with other more simple eukaryotic models including animal models, C. elegans and Drosophila (reviewed in [Epremyan et al., 2023], 11 references) significantly contributed in understanding Aβ and tau biology.”

“I recommend the authors to balance their work in the large amyloid field to recognize the utility of yeast models without overselling these methods, which have serious and not fully discussed limitations (the key one being the degree to which they recapitulate human pathology).”

Response: We added  “However, it should be noted, that yeast, as the unicellular organism fails as a model to study the multicellularity and cell–cell interactions, particularly important in the neuronal cross-talk that is of major importance to neurodegeneration Yeasts lack neuron-specific morphological structures, such as dendrites, axons and synapses. Consequently, the underlying neuron-specific molecular inventories are missing. Therefore, disease-associated processes uncovered in yeasts must be validated in neuronal model systems and eventually in human studies [45]” (lines 747-753).

We would like to draw the attention of the reviewer to the important paragraph in the review: “A great deal of research has been devoted to attempts to prevent the formation of Aβ aggregates or modulating and reducing Aβ toxicity. However, as noted above, there is no correlation between amyloid pathology and clinical manifestations such as cognitive decline [198]. Moreover, although amyloid pathology is developing well before AD symptom onset, it is not the earliest characteristic of AD. Neurons, post-mitotic and excitable cells have high energy needs accommodated almost exclusively by the mitochondrial oxidative phosphorylation system to meet their energy demands. Mitochondria play a crucial role in maintaining synaptic function. Energy deficiency and high level of mitochondrial division (fragmentation) are important early events promoting synaptic deficiency and neural cell death in AD [199,200].”

We were not the first to draw attention to this important fact. But we, using “normal” yeast cells, continued these studies, showing that 1) excess mitochondrial fragmentation induced by oxidative stress (the addition of prooxidant), is prevented and even reversed by mitochondrial-directed antioxidants, which is of the mitochondrial ROS as inducers of fragmentation (Goleva et al., SkQThy, a novel and promising mitochondria-targeted antioxidant. Mitochondrion. 2019 Nov;49:206-216. doi: 10.1016/j.mito.2019.09.001.); 2) using 100 individual giant yeast cells, we traced the spread of oxidative stress in each cell and showed that oxidative stress initially develops only in mitochondria, with mitochondrial fragmentation occurring, inhibited by mitochondrial-directed antioxidants, and both processes have always preceded the development of the generalized oxidative stress in the whole cell (Rogov et al., Propagation of Mitochondria-Derived Reactive Oxygen Species within the Dipodascus magnusii Cells. Antioxidants (Basel). 2021 Jan 15;10(1):120. doi: 10.3390/antiox10010120.). So, we’ve solved a problem that hasn’t been solved in 15 years with cardiomyocytes. Therefore, the yeast cannot be overestimated, but cannot be underestimated.

Especially when, according to an increasing number of scientists, the amyloid era, which views the formation of amyloid proteins as a leading factor in AD pathology, is giving way to the mitochondrial dysfunction era. In general, the role of yeast in detecting the earliest manifestations of violations in the various cellular networks and at the level of omics will only increase, due to their advantages over other models described in the review. Of course, yeast is something that research should only begin to accelerate testing of found potential therapeutic agents on more complex models, ultimately human, as we write in the final chapter of the review.

Reviewer 5 Report

The manuscript titled "Alzheimer's Disease: Significant Benefit from the Yeast-Based Models" by Epremyan et al. provides an extensive examination of the utilization of yeast models for investigating the molecular mechanisms involved in the development and progression of Alzheimer's disease (AD). The manuscript offers a comprehensive review of the existing research in this field. However, it is important to note that late onset sporadic AD represents the majority of AD cases. Therefore, it would be valuable for the authors to address the limitations of yeast-based models in studying these cases, as well as discuss the potential application of yeast-based models in investigating the less explored GWAS hits associated with AD. This would further enhance the manuscript's contribution to our understanding of AD pathogenesis and potential avenues for future research.

Author Response

Reviewer 5

First of all, we would like to thank the referee for useful comments and advises and are ready to answer the questions and criticism raised.

“The manuscript titled "Alzheimer's Disease: Significant Benefit from the Yeast-Based Models" by Epremyan et al. provides an extensive examination of the utilization of yeast models for investigating the molecular mechanisms involved in the development and progression of Alzheimer's disease (AD). The manuscript offers a comprehensive review of the existing research in this field. However, it is important to note that late onset sporadic AD represents the majority of AD cases. Therefore, it would be valuable for the authors to address the limitations of yeast-based models in studying these cases, as well as discuss the potential application of yeast-based models in investigating the less explored GWAS hits associated with AD. This would further enhance the manuscript's contribution to our understanding of AD pathogenesis and potential avenues for future research.”

Response: We mentioned the GWAS method three times, adding at the first mention ”Genome-Wide Association Studies (GWAS), a well-recognized powerful method for finding genomic areas underlying observed phenotypic variation. GWAS study of AD reported 44 single-nucleotide polymorphisms (SNP) associated with the late-onset AD (Zhang Q. et al., "Risk prediction of late-onset Alzheimer’s disease implies an oligogenic architecture," Nature Communications, vol. 11, no. 1, p. 4799, 2020/September/23 2020)” (lines 70-73).

Then “Thus, these studies have shown that the identification of Aβ toxicity modifiers, their associated mechanisms, and the discovery of suitable therapeutic targets can be performed using yeast Aβ models used in conjunction with GWAS and/or SGA” (lines 445-448).

and “This suggests that the ability to modeling intracellular Aβ toxicity and disruption of mitochondrial functions makes yeast-based Aβ models an extremely valuable tool for studying AD. Also, in combination with GWAS studies, these models become an excellent platform for screening promising compounds that can modify Aβ cytotoxicity” (lines 481-485).

For yeast-based models, everything in the subjunctive, so in future research.

Round 2

Reviewer 4 Report

The authors have undertaken an appropriate revision. They addressed my points appropriately. 

Author Response

Reviewer 4.

“The authors have undertaken an appropriate revision. They addressed my points appropriately.”

Response: We appreciate the work you have done and are grateful for your comments and advice, which helped improve our manuscript.